# TAMA: Tool-Augmented Multimodal Agent for Procedural Activity Understanding

## Abstract

Procedural activity assistants potentially support humans in a variety of settings, from our daily lives, e.g., cooking or assembling flat-pack furniture, to professional situations, e.g., manufacturing or biological experiments. Despite its potential use cases, the system development tailored for such an assistant is still underexplored. In this paper, we propose a novel framework, called TAMA, a Tool-Augmented Multimodal Agent, for procedural activity understanding. TAMA enables interleaved multimodal reasoning by making use of multimedia-returning tools in a training-free setting. Our experimental result on the multimodal procedural QA dataset, ProMQA-Assembly, shows that our approach can improve the performance of vision-language models, especially GPT-5 and MiMo-VL. Furthermore, our ablation studies provide empirical support for the effectiveness of two features that characterize our framework, multimedia-returning tools and agentic flexible tool selection. We believe our proposed framework and experimental results facilitate the thinking with images paradigm for video and multimodal tasks, let alone the development of procedural activity assistants.

## 1 Introduction

Procedural activities are ubiquitous, spanning our daily lives and professional settings, such as cooking (Peddi et al., 2024), assembly (Sener et al., 2022), manufacturing (Schoonbeek et al., 2024), lab experiments (Yagi et al., 2025), and medical practice (Jang et al., 2023), among others. Assistant systems can democratize such activities by providing supportive guidance that makes them accessible to beginners. Advances in large language models (LLMs) and vision-language models (VLMs) have significantly enhanced performance on existing video understanding benchmarks through improved pretraining and posttraining. For further improvement, we combine the ideas of reasoning and agent to enable the "thinking with images" paradigm (Su et al., 2025) as an inference-time technique for procedural activity understanding.

Procedural activity understanding involves comprehending both the actual process, captured in the recording, and the expected process, described in textual or visual instructions, and aligning them to detect potential mismatches (Hasegawa et al., 2025b). This cross-modal alignment can be more tractable by decomposing the overall process into more manageable subtasks. For instance, suppose one asks the following question while assembling a flat-pack furniture, "Did I make any mistake before attaching this part?" When humans approach this question, they typically examine the situation one by one. First, check the instructions to determine when and how the part is supposed to be attached. Next, they review the actions in the video to identify any misalignments, e.g., skipped steps or incorrect step orders. By repeating these steps as needed, they eventually either flag an error or conclude that no error exists and respond to the question.

One naive, yet typical approach for such video-centric multimodal tasks with VLMs is to provide all information as one input, i.e., feed to a model the concatenation of a question, instructions, and sampled frames from a recording, and obtain a prediction in one inference. This simple formulation aligns well with traditional workflow approaches that predefine information processing paths, e.g., keyframe selection that selects only informative frames, followed by answer prediction (Ye et al., 2025). It also works well with recent techniques, like prompt engineering (Liu et al., 2023) or reasoning model Jaech et al. (2024), both of which scale the inference time by outputting additional thought tokens, preceding its answer generation. However, due to the nature of single-pass prediction, errors

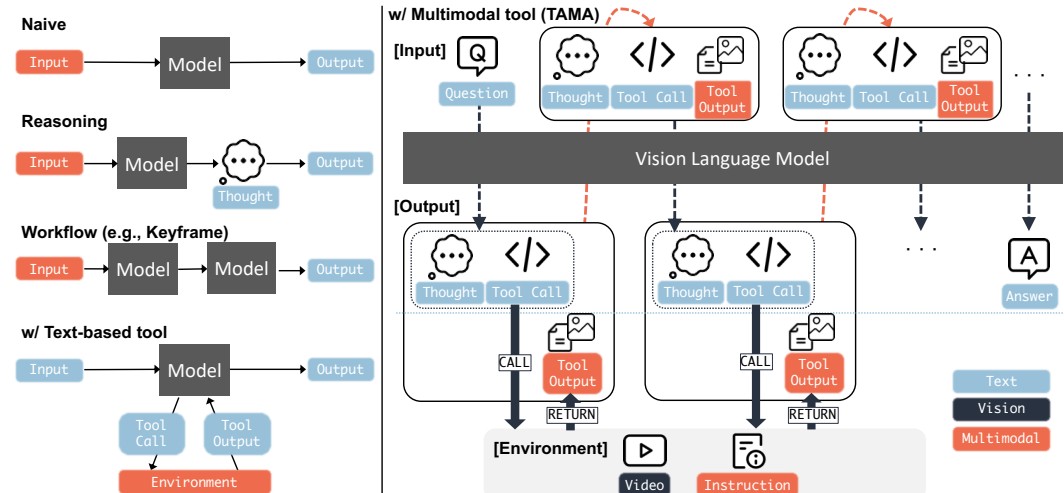

Figure 1: Left: Overview of existing approach. Right: Overview of our proposed approach, TAMA. Given a question as an initial input, a VLM-based agent generates its thought, followed by a tool call. Once a tool output is produced, the concatenation of the model output and the tool output is appended to the previous input to form the next input. Then, the model further generates either the next pair of a thought and tool call or an answer.

in beginning processes, e.g., frame selection, may be difficult to recover from, or managing context with multiple frames and many thought tokens poses long-dependency challenges for models (Sun et al., 2025).

Another growing direction is an agentic approach (Xi et al., 2025). Compared to the single-pass approaches, which are typically implemented with a predefined, fixed-step workflow, a language model (LM) as an agent answers a question by making use of the predefined information processes (tools) flexibly, proactively, and iteratively. Prior studies in video-centric tasks primarily design an agentic framework with text LLMs, which reason and decide actions, and semantic grounding tools, which provide textual conversion of visual information through captioning or OCR (Wang et al., 2024). In contrast, humans would perform interleaved reasoning and visual comprehension by using perceptual exploration tools, e.g., fast-forwarding videos by time stamps, zooming into one frame, changing camera angles, etc. While the paradigm of interleaved textual and visual reasoning, i.e., thinking with images (OpenAI, 2025a), has been gaining attention on image-centric tasks, its application to video-centric tasks, especially in training-free settings, is yet underexplored. Inspired by this gap, we pose the following research question: *Can VLMs make use of interleaved multimodal reasoning by using perceptual exploration tools to better perform video-centric multimodal tasks?*

In this work, we propose a novel training-free agentic framework, **TAMA** (**T**ool-**A**ugmented **M**ultimodal **A**gent), that enables interleaved multimodal reasoning by multimedia-returning tool use. Figure 1 illustrates the overview of our proposed framework. A VLM-based agent orchestrates tools that return either images or text to perceptually explore the current situation with its reasoning capability in an interleaved manner (§ 3). We experiment with our framework in a training-free setting, where VLMs are given only task and tool information as a prompt to see if current VLMs can make use of the tools out of the box. Our experimental results with both proprietary and open-weight models on ProMQA-Assembly (Hasegawa et al., 2025a) reveal that our framework can further elicit the performance for some models, i.e., GPT-5 (OpenAI, 2025b) and MiMo-VL (Xiaomi, 2025), although the performance change varies, as sometimes performance degrades under our framework, i.e., Qwen2.5-VL (Bai et al., 2025) and InternVL3 (Zhu et al., 2025). Yet, the result suggests that our framework can potentially bring out the models' capability in a zero-shot manner for video-centric multimodal tasks, considering that, while arguable, most of the tools from our experiments are likely to be unseen during training. As a behavioral analysis, we examined the tool-use patterns, aiming to provide potential reasons for performance discrepancies (§ 4). Furthermore, to provide the empirical evidence of our framework's efficacy, we conducted the ablation studies, w.r.t the modality of tool outputs, the flexibility of the tool selection, and the impact of frame sampling (§ 5).

In short, our contributions are threefold: (1) We propose a training-free agentic framework for interleaved multimodal reasoning with tools. (2) Our experiment shows that our framework can potentially improve the performance of VLMs. (3) Our ablation studies support that multimedia-return tools and agentic tool use are beneficial. We believe that our work stimulates research on the thinking with images paradigm for video understanding tasks, thus, more capable procedural activity assistants that benefit human society.

## 2 RELATED WORK

Our work is inspired by reasoning VLMs, video agents, and procedural activity understanding.

### 2.1 VISION-LANGUAGE MODEL

Vision-language models, which process visual and textual information together, have rapidly progressed over the past few years. Strong proprietary models are mostly VLMs by default (OpenAI, 2025b; Anthropic, 2025b; Google, 2025), and an increasing amount of competitive open/open-weights models have been released in the community (Bai et al., 2025; Zhu et al., 2025; Xiaomi, 2025). On top of the popular prompt techniques (Wei et al., 2022; Wang et al., 2023b), reasoning models are becoming dominant in public benchmarks (Jaech et al., 2024; Guo et al., 2025). While a reasoning paradigm is primarily on text, its variant, "thinking with images" (Su et al., 2025), has also been gaining attention. This paradigm introduces visual information into its textual thought process in an interleaved manner by making use of external tools (OpenAI, 2025a; Hu et al., 2024) or by using a native multimodal model that has the capability of synthesizing images as well (Team, 2024). Our work aligns with the former tool-driven thinking with images paradigm, specifically for video understanding tasks.

### 2.2 VIDEO AGENT

In video understanding studies, a traditional workflow system (Anthropic, 2025a), where a model processes data based on a fixed predefined order, has played a major role and is still competitive, due to its customizability, e.g., Socratic model (Zeng et al., 2022) or keyframe selection approaches (Ye et al., 2025; Arnab et al., 2025). In parallel to the general progress of VLMs and their agentic capability, agentic approaches are getting more attention in video tasks as well (Xi et al., 2025). An agent, typically a language model, flexibly and proactively selects an action based on tool descriptions and its thought process to understand the situation and answer a question. While prior agentic systems in video understanding tasks show their effectiveness, in most cases, their tools are for semantic grounding, i.e., converting images into text, with only a textual thought process (Wang et al., 2024; Tian et al., 2025). In contrast, our work features perceptual exploration tools, which help an agent to explore visually (Wu & Xie, 2024; Zhang et al., 2025b) and form interleaved multimodal reasoning. One concurrent work by Zhang et al. (2025a) also proposes to use frame sampling as a tool; however, their with-training and single-tool setup differs from our training-free and multi-tool setup.

### 2.3 PROCEDURAL ACTIVITY UNDERSTANDING

Procedural activity exists everywhere, where assistants can support users from their ego- and exocentric viewpoints by aligning observed actions in recordings with the expected actions in instructions. Due to the ubiquitous demands, prior studies have covered diverse domains: cooking (Stein & McKenna, 2013; Peddi et al., 2024; Lee et al., 2024), assembly (Ben-Shabat et al., 2021; Jang et al., 2019), manufacturing (Ragusa et al., 2021; Wang et al., 2023a; Schoonbeek et al., 2024), lab experiments (Yagi et al., 2025), and medical practice (Beyer-Berjot et al., 2016; Jang et al., 2023), among others (Haneji et al., 2024). While classification tasks are popular in those studies, some work explores other task formulations to facilitate the development of systems with more human-friendly and detailed responses. For instance, the ProMQA series proposes multimodal QA datasets on procedural activities, i.e., cooking and assembly (Hasegawa et al., 2025b;a). In this work, we adopt ProMQA-Assembly as our evaluation dataset, considering the instruction variety, i.e., target assembly image, in addition to both textual and image instructions (Example in Table 3). We leave it to future work to apply our method to ProMQA(-cooking) or other datasets.

Table 1: Our tool set.

| Function | Description | Example |
|---|---|---|
| Sample frame | Return sampled frames in the specified range at equal intervals from the specified angle of the camera. | `sample_frame(`
`    start='0:10', end='0:20', angle='center'`
`)` |
| Zoom in | Return the specified frame's cropped image based on the specified normalized bounding box. | `zoom_in(`
`    frame_id=106,`
`    bounding_box=[0.3, 0.4, 0.7, 0.8]`
`)` |
| Check instruction | Return an instruction in either text (DOT format[1]) or image (directed acyclic graph). | `check_instruction(mode='text')` |
| Check final picture | Return the target assembly image with parts. | `check_final_picture()` |

## 3 APPROACH

TAMA is a training-free agentic framework that enables interleaved multimodal reasoning by tool use. Before introducing our approach, we first define our target task, followed by the existing approaches. All approaches, including TAMA, are illustrated on Figure 1.

### 3.1 TASK FORMULATION

Our target task is multimodal question answering, specifically for understanding procedural activities. The input consists of: (1) a user's question in text, (2) a video recording of the activity up to the point when the question is asked, and (3) instructions provided in both image and text formats, including a target assembly image. The output is a textual answer.

### 3.2 EXISTING APPROACH

**Naive and Reasoning**  One prevalent approach with VLMs feeds the concatenation of sampled frames from a video, instructions, and a question into models (*naive*) (Fu et al., 2025). On top of this naive approach, prompt techniques or reasoning models are used to further enhance the performance (*reasoning*). While simple, depending on a model's valid context length, a model may not keep attending enough attention to initial frames in its decoding time (Sun et al., 2025).

**Workflow**  Most traditional studies can be categorized into *workflow*, where processes, e.g., LLMs and tools, follow a predefined sequential path. For instance, keyframe selection approaches can be seen as workflow systems when you treat the first stage of keyframe selection and the second stage of answer generation as two fixed-order processes/tools (Arnab et al., 2025). While customizable, since the process path needs to be predefined, careful path design would be required (§ 5.2).

**Agent with text-returning tool**  Arguably, due to the success of text LMs, this has been the major approach for existing agentic work for video understanding tasks: An agent, i.e., a text-only LM, devises an answer in response to a query/question by flexibly making use of tools that return text. When a tool is invoked, it accesses the environment for a textual instruction or a video file. When the target is text, the information is passed through the tool and returned to the agent. When the target is visual content, a tool, typically VLMs or task-specific models, performs semantic grounding by converting it into text, e.g., captioning, and returns it to the agent. While this approach can benefit from the evolving agentic capability of text-only LMs, vision-to-text conversion can be an information bottleneck, which may impair performance (§ 5.1).

### 3.3 OURS: TAMA

Our approach employs a VLM rather than a text-only LM as its agent and relies on multimedia-returning tools that return information in original modalities, i.e., text remains text and images remain images. Existing agent frameworks have proposed to integrate VLMs for video understanding tasks, yet mainly as tools, rather than agents (Yang et al., 2023; Tian et al., 2025). Motivated by the success of GUI agents (Zhang et al., 2024), we propose to use VLMs as agents for video understanding tasks so that an agent can reason and call tools based on original multimodal information. To leverage the

Table 2: Result.

| Model | Naive | Reasoning | TCoT | TAMA (ours) |
|---|---|---|---|---|
| GPT-5 mini | 58.1 | 56.9 | 58.8 | **63.7** |
| GPT-5 | 58.7 | 60.0 | 57.9 | **67.0** |
| Claude 4 Sonnet | 46.4 | **56.8** | 52.0 | 55.6 |
| Gemini 2.5 Flash | 41.6 | 48.8 | **54.9** | 52.4 |
| Qwen2.5-VL 32B | 44.0 | **44.6** | 40.8 | 44.0 |
| InternVL3 38B | **50.5** | 48.2 | 48.5 | 46.3 |
| MiMo-VL 7B | 33.1 | 46.4 | 46.8 | **49.6** |

capability of VLM-based agents, we define four tools, as summarized in Table 1. `sample_frame` and `zoom_in` enable an agent to explore a video at different granularities. `check_instruction` and `check_final_picture` help an agent to access manuals in different modalities. Essentially, the tools are defined so that models can explore information perceptually, rather than ground visual information in text, to prevent any information loss during information conversion. Tools are all implemented as Python functions that access local files. We explore this framework in a training-free setting to investigate current VLMs' zero-shot capability. As illustrated in Figure 1, we feed a prompt with a question (and tool information) to a model and generate a tool call with a thought process. Once we obtain a tool output by executing the tool locally, we append both the model output and the tool output to the previous input, which is again fed to a model.

## 4 EXPERIMENT

We compare TAMA against existing approaches on a multimodal QA task to verify its effectiveness.

### 4.1 BASELINE APPROACH

We first compare our framework with three baseline approaches: naive, reasoning, and workflow. For the naive and reasoning approaches, we feed the concatenation of sampled frames, an instruction (text), a target assembly image, and a question as one input, and obtain an answer, preceded by a thought process for the reasoning. For workflow, we experiment with Temporal Chain-of-Thought (TCoT) (Arnab et al., 2025), a two-stage approach, where VLMs select keyframes based on each question, and the same model answers it based on the selected frames. We chose TCoT as our baseline because it is also a training-free approach. As for the text tool-based agentic approach, we conduct an ablation study to compare text tools and multimedia tools in § 5.1. As all approaches, including ours, are model-agnostic, we apply these approaches to the following models.

### 4.2 EXPERIMENTAL SETUP

In our experiment, we include both proprietary and open-weight models. For proprietary models, we chose GPT-5, GPT-5 mini, Claude 4 Sonnet, and Gemini 2.5 Flash, based on their performance on public benchmarks and costs. For open-weight models, among VLMs, we selected three models based on their reported capabilities on agentic benchmarks and also computational demands: Qwen2.5-VL 32B, InternVL3 38B, and MiMo-VL 7B. To achieve TAMA's interleaved thought process, we use either reasoning mode for proprietary models and MiMo-VL 7B, or ReAct-style prompting (Yao et al., 2023) with zero-shot CoT (Kojima et al., 2022) for Qwen2.5-VL 32B and InternVL3 38B. We format our iterative thought-call-return process in a similar way to multi-turn conversations, where we set minimum and maximum turns as hyperparameters. In case a model outputs an answer too quickly or too late, we include a cut-in message to encourage the model to think more or answer in the next turn. All experiments are done without any in-context examples, i.e., zero-shot inference.

As our evaluation dataset, we use ProMQA-Assembly, a multimodal QA dataset for procedural activity understanding, which has a unique setting of including video recording, instructions, a target assembly image, and a question as input. Following the prior work, we adopt the LLM-as-a-judge for assessing the quality of predictions. A judge model outputs the score, 0 (incorrect), 1 (partially correct), or 2 (correct), and we take the average with scaling to 0 to 100 by multiplying by 50. All numbers are reported by a single run of experiments. More details are available in Appendix A.

Table 3: Example with an instruction image (top left), target assembly image with parts (top right), sampled frames from a recording (middle), and a pair of a question and ground-truth answers, followed by GPT-5's responses from each approach.

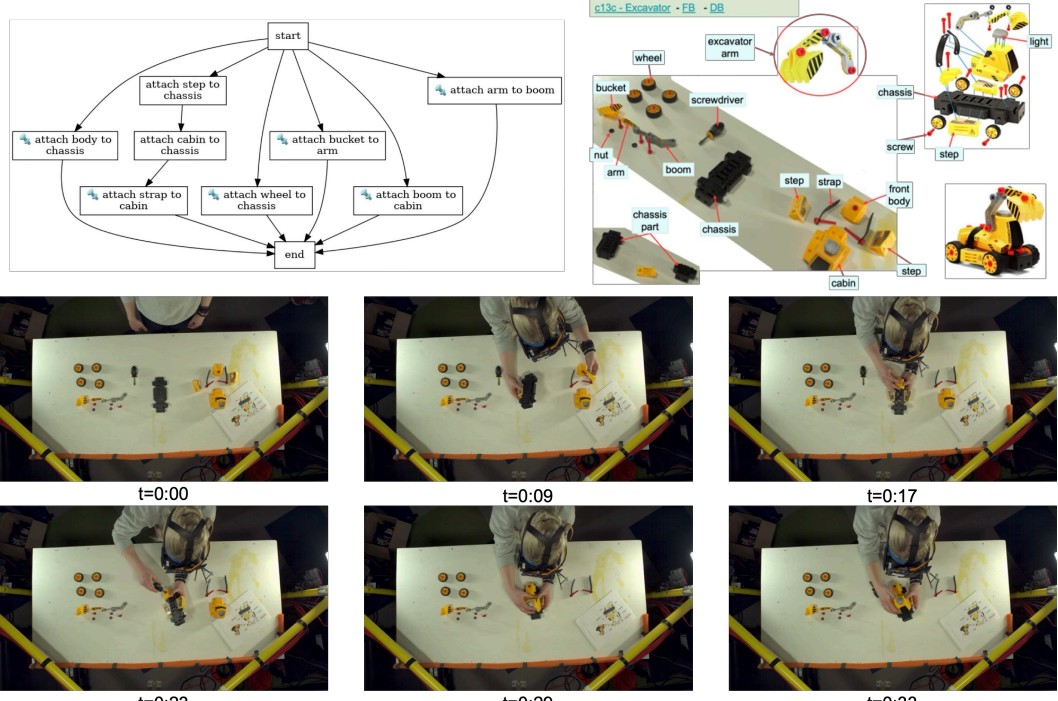

| | |
|---|---|
| Question: | What should I have done differently when attaching the body to the chassis? |
| Ground Truth: | Nothing, everything is going great! / There was nothing you needed to do differently. |

| | |
|---|---|
| Naive: | ✗ You should have secured the body to the chassis with a screw instead of just snapping it on. |
| Reasoning: | ... Maybe they should have used a different orientation for the screw, ... 
 ✗ You should have secured the body to the chassis with a screw. |
| TCoT: | {"selected_frames": [...], "justifications": "These frames show the actual step of attaching the yellow body onto the black chassis, including how it's oriented and aligned. ..."}, ... The instructions clearly state to use a screw for this step. ... the answer points towards using a screw ... 
 ✗ Secure the body to the chassis with a screw—use the screwdriver to fasten it instead of just placing it on. |
| TAMA: | ... I think I can call the functions to get the needed details. <tool=sample_frame> <tool_output=frames> There could be several issues: wrong orientation, incorrect screws, or missing steps like inserting the window piece first. ... It might be helpful to check the instructions ... <tool=check_instruction> <tool_output=instruction> ... call check_final_picture to understand the correct orientation. ...<tool=check_finale_picture> <tool_output=image> ... The answer likely involves checking the orientation and ensuring the correct number of screws align with the holes. ... Using zoom tools on specific frames might help clarify things ... <tool=zoom_in> <tool_output=image> ... I'll call for a zoom-in again ... <tool=zoom_in> <tool_output=image> ... I want to zoom in ... I'll request a different bounding box ...<tool=zoom_in> <tool_output=image> ... Despite some ambiguity from the video, ... they did it correctly ... 
 ✓ Nothing—you aligned the front body correctly on the chassis and secured it with a screw as the instructions require. |

## 4.3 RESULT AND DISCUSSION

Table 2 shows the experimental result. Among all the combinations, GPT-5 with our framework exhibits the best performance. For GPT-5, GPT-5 mini, and MiMo-VL 7B, TAMA outperforms other approaches for each model, e.g., 14.1% improvement from the naive approach to TAMA by GPT-5. Gemini 2.5 Flash with TAMA shows superior performance over naive and reasoning approaches, but lags behind TCoT. Claude 4 Sonnet prefers our framework over naive and TCoT, but its text-only reasoning process shows slightly better performance than ours. For Qwen2.5-VL 32B and InternVL3 38B, neither TAMA nor TCoT outperforms the naive or reasoning approaches.

Table 4: Analysis of TAMA.

| Model | #frames (avg./median) | Tool Frequency per Question | | | | | #turn | |
|---|---|---|---|---|---|---|---|---|
| | | sample | zoom | inst. | pic. | total | 1st ans. | total |
| GPT-5 mini | 20.8 / 20.0 | 1.2 | 0.4 | 1.1 | 1.0 | 3.7 | 3.1 | 8.0 |
| GPT-5 | 24.2 / 21.0 | 1.2 | 1.8 | 1.2 | 1.0 | 5.2 | 4.0 | 8.8 |
| Claude 4 Sonnet | 31.8 / 26.0 | 1.9 | 0.9 | 1.6 | 1.1 | 5.4 | 3.2 | 9.0 |
| Gemini 2.5 Flash | 12.7 / 7.0 | 1.1 | 0.4 | 1.5 | 0.6 | 3.6 | 5.1 | 7.3 |
| Qwen2.5-VL 32B | 22.5 / 26.0 | 1.2 | 0.2 | 0.9 | 0.8 | 3.1 | 4.7 | 9.3 |
| InternVL3 38B | 18.0 / 14.0 | 1.6 | 0.3 | 1.1 | 1.0 | 3.9 | 4.3 | 9.9 |
| MiMo-VL 7B | 14.1 / 11.0 | 1.3 | 0.2 | 0.9 | 0.8 | 3.2 | 3.9 | 9.2 |

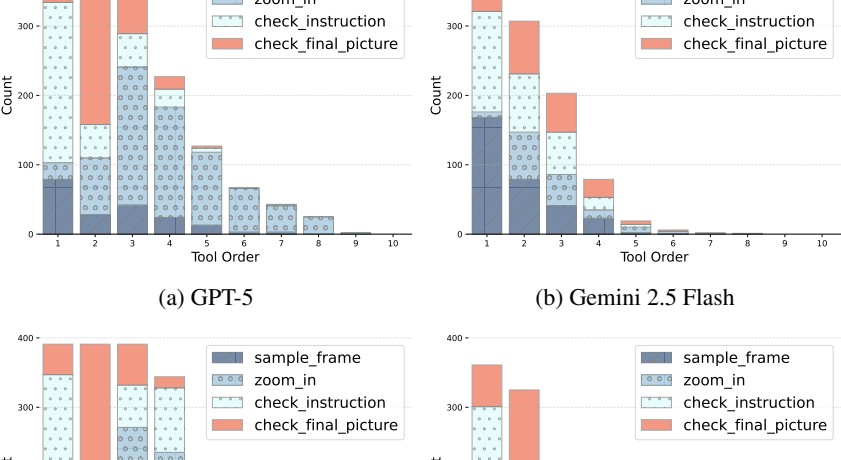

(a) GPT-5      (b) Gemini 2.5 Flash

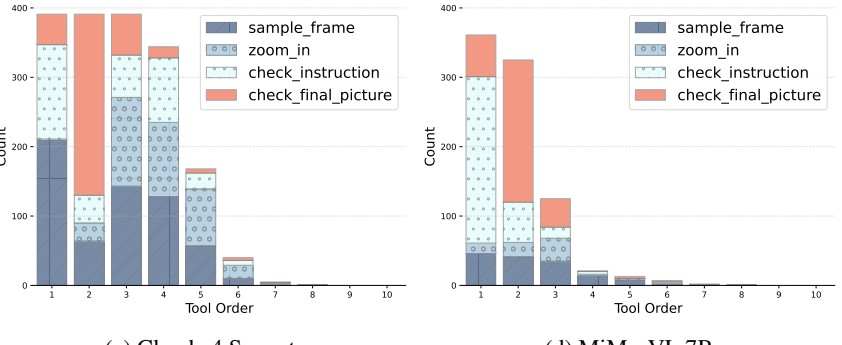

(c) Claude 4 Sonnet      (d) MiMo-VL 7B

Figure 2: Tool usage pattern.

To better understand model differences, we investigated several aspects of each model's output: the total number of sampled frames, tool usage frequency, and the number of turns for initial and final answers per question (Table 4). We also examined tool usage patterns (Figure 2). Gemini 2.5 Flash sampled a substantially smaller number of frames, which can be a potential reason for its less performant result with TAMA (See § 5.3 for our empirical support). As the Gemini API documentation[2] describes that it can specify points in a video by a timestamp, the model is expected to be familiar with timestamps. Thus, since the model calls `sampel_frame` in a similar frequency to other models, it may tend to select fewer frames, as reported in the TCoT paper. In contrast, Claude 4 Sonnet sampled the largest number of frames among all models, even though the model does not benefit from our framework. This suggests that the number of frames itself does not correlate with the effectiveness of our framework. GPT-5 and Claude 4 Sonnet call tools more frequently than others, where GPT-5 notably prefers the zoom-in tool, as highlighted in the tool pattern figures. This indicates that, in conjunction with its superior performance, GPT-5 may be specifically trained for the thinking with images paradigm with similar tools, and the capability may be transferable to video understanding tasks under our framework. Table 3 shows one set of example outputs from GPT-5. The model uses `zoom_in` tools in the latter half of the process to be more certain of its answer.

---

[2] https://ai.google.dev/gemini-api/docs/video-understanding

Table 5: Perf. w/ Text vs Multimedia tool.

| Model | Text | Multi |
|---|---|---|
| GPT-5 mini | 59.0 | 63.7 |
| Gemini 2.5 Flash | 48.2 | 52.4 |
| Qwen2.5-VL 32B | 39.0 | 42.1 |
| MiMo-VL 7B | 50.9 | 49.6 |

Table 6: Perf. w/ and w/o presample.

| Model | TAMA | TAMA w/ presample |
|---|---|---|
| GPT-5 mini | 63.7 | 63.2 |
| Gemini 2.5 Flash | 52.4 | 55.0 |
| Qwen2.5-VL 32B | 44.0 | 49.0 |
| InternVL3 38B | 46.3 | 46.7 |
| MiMo-VL 7B | 49.6 | 49.1 |

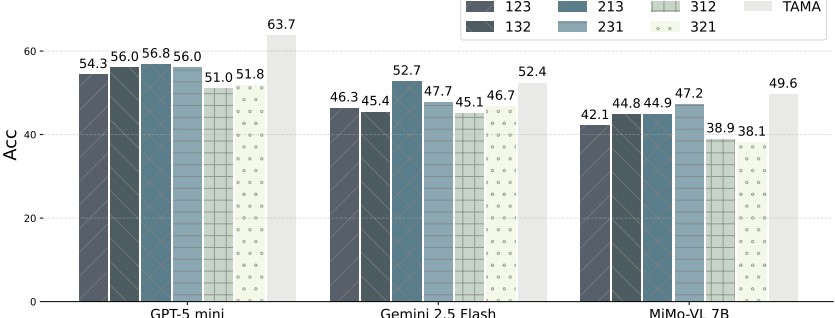

Figure 3: Performance of workflow vs agentic approach (TAMA). Each number represents one tool operation in the workflow approach: "1" is the uniform sampling, "2" is the instruction check, and "3" is the target assembly image check, and each digit sequence defines the execution order of the tools.

Qwen2.5-VL 32B and InternVL3 38B show similar characteristics to GPT-5 mini, in terms of the number of frames and tool frequency (Figure 13 in Appendix). However, during our manual inspection, we noticed that these open-weight models sometimes failed to follow the intended ReAct-style prompting. While we expected an interleaved thought process, the models occasionally refused to output any reasoning and instead produced only tool calls one after another. This suggests that these models would require additional tuning to be applied in our framework. MiMo-VL 7B does not show any particular uniqueness in its tool frequency or pattern, while it is the only open-weight model that benefits from our framework. Based on the claims in the MiMo-VL paper and its result (naive < reasoning < TAMA) in our experiment, one can guess that the capability of textual reasoning may be related to interleaved multimodal reasoning. However, as the result of Claude 4 Sonnet may refute (naive < TAMA ≤ reasoning), further investigation would be needed to understand what training contributes to interleaved multimodal reasoning, and we leave it for future work.

## 5 ABLATION STUDY

Our proposed framework, TAMA, distinguishes itself from prior work in two aspects: multimedia-return tools and agentic, flexible tool selections. To further understand their effects, we conducted the two ablation studies. In addition, we experimented with one heuristic strategy, presampling, inspired by the undersampling behavior of Gemini 2.5 Flash.

### 5.1 TEXT-RETURN TOOL VS MULTIMEDIA-RETURN TOOL

The first characteristic lies in tools capable of returning multimedia outputs. Given a tool call, our tools can return either text or images, contrary to the text-returning tools. As mentioned in § 3.2, we conducted a controlled experiment by defining a semantic-grounding version of our perceptual exploration tools. Specifically, image-returning tools are instead returning captions of images, where captions are obtained by prompting the same model as its agent model. To isolate the effect of agent models, we use VLMs for both text-returning and multimedia-returning tools, instead of text LMs, which are typical for agents with text-returning tools. GPT-5 mini, Gemini 2.5 Flash, Qwen2.5-VL 32B, and MiMo-VL 7B are used in this experiment. According to the result in Table 5, GPT-5 mini, Gemini 2.5 Flash, and Qwen2.5-VL 32B with multimedia-returning tools outperform those with

text-returning tools, while MiMo-VL 7B prefers text-returning tools. One possible reason for the MiMo-VL's preference may stem from its video re-captioning pipeline for pretraining, where they produced dense, fine-grained captions for each video. However, overall, our experiment shows a positive impact of multimedia-returning tools.

## 5.2 WORKFLOW VS AGENTIC TOOL USE

Secondly, we investigate the effect of its proactive and flexible tool selection. Specifically, we compared TAMA with a fixed-order workflow approach. We selected the following three tools with fixing arguments: namely, `sample_frame` with uniform sampling from each recording, `check_instruction` with text mode, and `check_final_picture`. The outputs of these tools are fed to a model sequentially, while the model is prompted to output only its thought process without any tool calls. Once all information is given, a model is instructed to produce an answer. In this experiment, we included all the permutations of these three operations (6 orders in total) using GPT-5 mini, Gemini 2.5 Flash, and MiMo-VL 7B. Figure 3 summarizes the result. We found that all permutations of the workflow approach degraded the performance, regardless of tool orders, except for one combination. When Gemini 2.5 Flash received information in the order of textual instructions, sampled frames, and the target assembly image, it performed comparably to TAMA. These results demonstrate the superior performance and cost-effectiveness of the agentic approach compared to the workflow-based method. Although the workflow approach can be tuned to match the agentic approach's performance, the agentic approach demonstrates superior usability. It achieves comparable or better performance without tuning by flexibly selecting appropriate tools and execution orders for each question, making it more efficient and user-friendly.

## 5.3 PRESAMPLING

As we found in our investigation (§ 4.3), some models, i.e., Gemini 2.5 Flash and MiMo-VL 7B, tend to select fewer frames than others. Arnab et al. (2025) addresses this point by compensating with uniformly sampled frames in their TCoT approach. Inspired by this, we also hypothesize that feeding additional frames may benefit those models. Specifically, we append the uniformly sampled frames from each recording to the initial prompt, which consists of a question and task information. This can be thought of as a hybrid approach of workflow and agentic framework. We conducted this experiment to get a better sense of which tool selection capabilities would be beneficial to incorporate into future training for video understanding tasks, with a specific focus on sampling. We primarily targeted Gemini 2.5 Flash, InternVL3 38B, and MiMo-VL 7B, as they had fewer sampled frames. We also included GPT-5 mini and Qwen2.5-VL 32B for comparison. As shown in Table 6, Gemini 2.5 Flash gains the benefit from this presampled strategy, while the performance of InternVL3 38B and MiMo-VL 7B did not change. Contrary to our expectation, Qwen2.5-VL 32B improves its performance with this strategy, although the number of its sampled frames is around the average of other models. While some models have already shown their capability of making use of our framework, this presampling experiment implies that additional training with respect to sampling may benefit these models.

## 6 CONCLUSION

In this work, we propose a novel training-free agentic framework, TAMA, to enable interleaved multimodal reasoning with tool use. Our experimental result shows that our framework for the thinking with images paradigm improves the performance of models such as GPT-5, GPT-5 mini, and MiMo-VL 7B. While some other models, Gemini 2.5 Flash and Qwen2.5-VL 32B, show their potential with the hybrid approach with presampling, the other models, e.g., Claude 4 Sonnet or InternVL3 38B, do not gain benefits, arguably because they are not familiar with an interleaved reasoning process or zero-shot use of our tools. Yet, together with the ablation study results on multimedia-returning tools and agentic tool selection, our work provides empirical support for our zero-shot, agentic prompting technique in a multi-turn setting. We believe that our work can facilitate the research on the perceptual exploration tools and interleaved multimodal reasoning for video understanding tasks, let alone the development of procedural activity assistants that benefit human society.

ETHICS STATEMENT

Our work does not introduce any training data, which may introduce additional biases or harmful content to VLMs. However, the negative contents inherent in VLMs from pretraining or posttraining may emerge within our framework. If our framework is to be deployed for production, rigorous evaluation against biases, fairness, privacy, jailbreak, etc, needs to be performed on top of our performance-focused evaluation, including the thought process.

REPRODUCIBILITY STATEMENT

We provide the general description of our proposed approach in § 3.3 and the experimental setup in § 4.2, which is further detailed in Appendix A. We also provide the prompt templates for our experiment in Appendix B. Furthermore, we attach the anonymized code used in our experiments as a supplemental material.

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

## A    EXPERIMENT DETAILS

We share the further details of our experiments in this section.

### A.1    TCoT IMPLEMENTATION

TCoT consists of two stages: the first stage is frame selection, and the second is answer generation. We used the dynamic-segment TCoT, where each input video is split into a fixed number of $l$ segments and each segment is fed to a model that generates the indices of frames for the second answer generation stage. Given the maximum number of frames, $k$, in each inference, if more than $k$ frames exist in one segment, $k$ frames are sampled from each segment. Once frames are selected from each segment, they are concatenated to form an input for answer generation. When the number of frames in this concatenation is more than $m$, $m$ frames are uniformly sampled. In addition to the selected frames in the first stage, they add uniformly sampled $u$ frames for temporal coverage. Thus, the input of the second stage consists of at most $m + u$ frames, which is fed to a model with a question and instructions to generate an answer. Hyperparameters are set as follows: $l = 4$, $k = 32$, $m = 48$, and $u = 16$. Prompt templates are available in Figure 4 and 5.

### A.2    TAMA IMPLEMENTATION

As described in § 4.2, we format our interleaved multimodal reasoning processes as multi-turn conversations. To put it simply, an input consists of [`system prompt`, `user question`, `model thought`, `model tool call`, `tool output`, `model thought`, `model tool call`, `tool output`, ... ]. However, API specifications of any proprietary models allow this format as is, i.e., either tool outputs cannot include images (OpenAI) or tool outputs need to be included in `user` messages (Anthropic and Google). Under this restriction, we, instead, add a note of "Asking a user to provide tool outputs." as tool outputs and add actual tool outputs with images in `user` messages. When we spot a case where a model does not generate an answer after $i$ turns or a case where a model generates an answer before $j$ turns, we include a cut-in `user` message to either encourage the model to answer or use more tools. We set the maximum number of turns as $h$, and we stop the iteration regardless of whether or not an answer is generated. The maximum number of frames that `sample_frame` returns is $k$, and the maximum number of frames in an input is $n$. If more than $k$ frames are selected, we pick $k$ frames at equal intervals. If more than $n$ images are included in one prompt, we remove the beginning images until the total number of images is equal to $n$. Figure 10 contains the detailed definition of our tools in the YAML format (Figure 11 for the text version). At most one tool is executed, even when multiple tool calls are generated. When a model outputs multiple tools in one output, we simply pick the first one to execute. Hyperparameters are set as follows: $i = 5$, $j = 2$, $k = 32$, $n = 64$, $h = 10$, $i = 8$, and $j = 5$. Prompt templates are available in Figure 6 and 9.

### A.3    MODEL SELECTION

Our model selection is mainly based on the performance and capability of a model, under the constraints of our cost budget and academic computational resources. The following are the reasons for other possible models we did not include in our experiments. Gemini 2.5 Pro returns server-side errors insufferably frequently at the time we experimented, so we ended up not using it, although its estimated cost is around the same as GPT-5 or Claude 4 Sonnet. We did not experiment with Claude 4/4.1 Opus due to their high costs. We did not use Qwen2.5-VL 72B due to the suspicion of its bug related to tool use, more specifically, it outputs a strange character every time it outputs tool calls. Following the size of Qwen2.5-VL, we used InternVL3 38B, instead of InternVL3 78B. GLM-4.5V (Hong et al., 2025) was not included because it did not fit into the 4 A6000 GPUs. Qwen3-VL [3] came out two days before the deadline of this submission, and we did not include it in our experiments.

The model IDs used in our experiments are as follows: `gpt-5-mini-2025-08-07` (GPT-5 mini), `gpt-5-2025-08-07` (GPT-5), `claude-sonnet-4-20250514` (Claude 4 Sonnet), `gemini-2.5-flash` (Gemini 2.5 Flash), `Qwen/Qwen2.5-VL-32B-Instruct`

---

[3] `https://huggingface.co/Qwen/Qwen3-VL-235B-A22B-Instruct`

Table 7: API Cost (USD)

| Model | Naive | Reasoning | TCoT | TAMA |
|---|---|---|---|---|
| GPT-5 mini | 1.6 | 0.81 | 5.7 | 4.2 |
| GPT-5 | 7.6 | 4.8 | 41 | 40 |
| Claude 4 Sonnet | 10 | 13 | 63 | 41 |
| Gemini 2.5 Flash | 0.86 | 2.3 | 11 | 5.1 |

(Qwen2.5-VL 32B), `OpenGVLab/InternVL3-38B` (InternVL3 38B), and `XiaomiMiMo/MiMo-VL-7B-RL-2508` (MiMo-VL 7B).

## A.4 OTHER DETAILS

The naive and reasoning approaches receive 32 uniformly sampled frames in their inputs. API services sometimes show their instability, returning server-side errors. In such cases, we run a model one more time to see if we can obtain a result. When we do not obtain results after attempting twice, we just include None as an answer. To access models, we use APIs for proprietary models and we run locally for open-weight models with the server mode of the `vllm` library. For reasoning, we set either "medium" or 2048 for reasoning effort/budget, and 512 as the maximum number of output tokens. Images are all scaled to the resolution of $640 \times 360$, and we use the center angle for recordings, unless specified. For Qwen2.5-VL 32B, we used our custom chat template because the original one from the HuggingFace Hub does not contain the templates for tool use. For InternVL3 38B, we used a previous version of its chat template because the latest one does not include the templates for tool use. For each inference of open-weight models, we used at most 4 A6000 GPUs (48GB memory) throughout our experiments. Table 7 shows the reference costs. Each cost represents the total cost of one model's experiment, i.e., obtaining answers for all questions in the evaluation dataset.

## B PROMPT TEMPLATE

```
You will be given a question about a video, following frames from the
    video.
Question: {question}
Return the frame ids which can answer the given question.
Please use the following JSON format for your output:
{
    "frame_ids": [List of integer/frame IDs],
    "justification": "<justification about your output>"
}
```

Figure 4: Prompt for frame selection in TCoT.

```
Frames: {frames}
Parts: {target assembly image}
Instruction: {dot}
An instruction is represented as a directed, acyclic partial graph, where
    a node is a step and a relation is the order of steps.
For instance, if there is a directed edge between node A and node B (A ->
    B), A needs to be done before B is performed.
You will be given a question about a video. You are provided frames from
    the video, retrieved by an intelligent agent. You are also provided
    with instructions and parts image.
It is crucial that you imagine the visual scene as vividly as possible to
    enhance the accuracy of your response. Answer in the following
    format: <answer>your answer</answer>
Question: {question}
```

Figure 5: Prompt for answer generation in TCoT.

```
You are helping a user performing an toy assembly task by checking their
    activity recording.

You have tools/functions to access the following information:
- video/recording of the activity
- instructions/manuals for the toy
- final picture image of the toy
When you get a question, call the tools to understand the user's current
    situation so that you can answer the question confidently.
When you finish analyzing the given information, make sure to answer the
    question in the following format:
<answer>your answer</answer>

Note:
- Each question is asked at the end timing of its recording. So make sure
     to contextualize each question in the recordings.
- An answer should be one, or a few concise sentence(s).
- Tools can be called multiple times until you obtain enough evidence to
    answer
the question confidently.
- After each tool call, make sure to think if the returned output is
useful/sufficient for answering the question.
- Each tool can be called multiple times, but tools can be called one at
    a time.
```

Figure 6: System prompt for TAMA

```
You are helping a user perform a toy assembly task.

You have tools/functions to access the following information:
- video/recording of the activity in text/caption
- instructions/manuals for the toy in text
- final picture image of the toy in text/caption
When you get a question, call the tools to understand the user's current
    situation so that you can answer the question confidently.
When you finish analyzing the given information, make sure to answer the
    question in the following format:
<answer>your answer</answer>

Note:
- Each question is asked at the end timing of its recording. So make sure
     to contextualize each question in the recordings.
- An answer should be one, or a few concise sentence(s).
- Tools can be called multiple times until you obtain enough evidence to
    answer
the question confidently.
- After each tool call, make sure to think if the returned output is
    useful/sufficient for answering the question.
- Each tool can be called multiple times, but tools can be called one at
    a time.
```

Figure 7: System prompt for TAMA (text)

## C    RESULT

Figure 13 shows the remaining models' tool-use patterns.

## D    LIMITATION

One limitation is in our experiment, regarding model variety. While we evaluated both proprietary
models and open-weight models, our selection may look small, considering the continuous stream

```
You are helping a user who is performing a toy assembly task by checking
    their activity recording.

As a starting point, you will be given a question.
Then, you will be given the following information one by one:
- video/recording of the activity
- instructions/manuals for the toy
- target assembly image of the toy
Once you receive all, make sure to answer the question in the following
    format:
<answer>your answer</answer>

Note:
- Each question is asked at the end of its recording. So make sure to
    contextualize each question in the recordings.
- An answer should be one or a few concise sentences.
- Make sure to think if the given information is useful/sufficient for
    answering the question.
```

Figure 8: System prompt for our workflow approach in § 5.2.

```
I have been working on the task for {duration}.
I have a question. <question>{question}</question>
```

Figure 9: Initial user prompt for TAMA

of model releases. In fact, only a few meet our requirements under our academic computational resources. Our framework requires a model to be a VLM that has agentic behavior/tool-use capability. Even when the paper/blog of a model mentions the benchmark numbers on agentic tasks, they may not always release "chat_template," which is crucial to render input information into their specific input format used in their training. If the templates are not available, we would need to come up with one by educated guesses, which may underrate their capabilities. Another limitation lies in the cost and efficiency of our framework. While we observed improved performance for some models, as our framework involves multiple inferences for each question, inference time gets longer with more computational cost, especially compared to the naive approach. Potential future directions to address this point involve shorter, yet higher-quality multimodal reasoning paths or distillation to smaller models. Additionally, the size of the evaluation data may hinder the comparison among models with small differences.

## E   LLM USAGE

We used LLM-powered AI services when drafting this paper, specifically for refining phrases or correcting grammatical errors, but not for ideation or more advanced purposes.

```yaml
- name: "sample_frame"
  description: |
    Function to sample frames in the video between the range with the rate.

    Output consists of a list of 1 fps sampled frame filepaths.
    Frame files are represented with their timestamps in second.
    The maximum number of frames is 30, and if more than the maximum
        number of frames are requested, the fps rate gets reduced to meet
        the requirement.
  args:
    start_time: {type: 'string', description: "The start time of the range
        to sample frames in the format of mm:ss."}
    end_time: {type: 'string', description: "The end time of the range to
        sample frames in the format of mm:ss."}
    angle: {type: "string", description: "camera angle of the video.",
        enum: ["center", "top", "right-bottom", "right-center", "right-top
        ", "left-bottom", "left-center", "left-top"]}

- name: "zoom_in"
  description: |
    Function to zoom in one frame.
    You can specify where to zoom-in by a normalized bounding box in the
        format of [x1,y1,x2,y2], where 0 < x1 < x2 < 1 and 0 < y1 < y2 < 1.

    (x1, y1) corresponds to the top left corner, and (x2,y2) corresponds
        to the bottom right coner.
  args:
    frame_id: {type: "integer", description: "the id of the frame to zoom-
        in"}
    angle: {type: 'string', description: "camera angle of the video", enum
        : ["center", "top", "right-bottom", "right-center", "right-top", "
        left-bottom", "left-center", "left-top"]}
    bounding_box: {type: "array", description: "normalized bounding box in
        the format of [x1,y1,x2,y2]", items: {type: "number"}}

- name: "check_instruction"
  description: |
    Function to access the instruction in text or image.
    An instruction is represented as a directed, acycle partial graph,
        where a node is a step and a relation is a order of steps.
    For instance, if there is a directed edge between node A and node B (A
        -> B), A needs to be done before B is performed.
    Instructions can be checked in either text or image:
    - text: instructions are represented as text in the DOT format.
    - image: instructions are represented as an figure of a graph.
  args:
    mode: {type: "string", description: "either text or image"}

- name: "check_final_picture"
  description: |
    Function to access the image of the final picture and parts of the
        target toy car.
    The image may contain its exploded view as well.
  args: null
```

Figure 10: TAMA's multimedia-returning tool definitions in the YAML format.

```yaml
- name: "sample_frame"
  description: |
    Function that returns a detailed description of sampled frames in the
        video between a specified range.
    Output consists of one description, based on the sampled frames.
    The default sample rate is 1 fps, and the maximum number of frames is
        30.
    If the specified range contains more than 30 frames, i.e., the range
        exceeds 30 seconds, the fps rate gets reduced so that the number
        of frames is less than or equal to 30.
  args:
    start_time: {type: 'string', description: "The start time of the range
        to sample frames in the format of mm:ss."}
    end_time: {type: 'string', description: "The end time of the range to
        sample frames in the format of mm:ss."}
    angle: {type: "string", description: "camera angle of the video.",
        enum: ["center", "top", "right-bottom", "right-center", "right-top
        ", "left-bottom", "left-center", "left-top"]}

- name: "zoom_in"
  description: |
    Function that crops one frame and returns the detailed description of
        the cropped frame.
    You can specify where to zoom-in by a normalized bounding box in the
        format of [x1,y1,x2,y2], where 0 < x1 < x2 < 1 and 0 < y1 < y2 < 1.

    (x1, y1) corresponds to the top left corner, and (x2,y2) corresponds
        to the bottom right coner.

  args:
    frame_id: {type: "integer", description: "the id of the frame to zoom-
        in"}
    angle: {type: 'string', description: "camera angle of the video", enum
        : ["center", "top", "right-bottom", "right-center", "right-top", "
        left-bottom", "left-center", "left-top"]}
    bounding_box: {type: "array", description: "normalized bounding box in
        the format of [x1,y1,x2,y2]", items: {type: "number"}}

- name: "check_instruction"
  description: |
    Function that returns the assembly instruction in text.
    An instruction is represented as a directed, acycle partial graph,
        where a node is a step and a relation is a order of steps.
    For instance, if there is a directed edge between node A and node B (A
         -> B), A needs to be done before B is performed.
    Instructions are represented as text in the DOT format.
  args: null

- name: "check_final_picture"
  description: |
    Function that returns the detailed description of the final picture,
        parts image, and possibly with an exploded view as well.
  args: null
```

Figure 11: TAMA's text-returning tool definitions in the YAML format.

```
## Instruction ##
This is an evaluation task.
You will be given a question, gold answer(s), and predicted answer.
Your task is to evaluate if the predicted answer matches against the gold
    answer(s).

Here is/are the step(s) they have already performed in the actual order:
{previous_steps}

Give your ternary judge 0, 1, or 2:
* 0 means the predicted answer is wrong (unmatch)
* 1 means the predicted answer is partially correct/wrong (partial match)
* 2 means the predicted answer is correct (match)
When multiple gold answers are available (provided as a list), the
    predicted answer is correct/partially correct if it matches/partially
     matches with at least one of the gold answers.

Provide your feedback as follows:
## Feedback ##
[Rationale] (your rationale for the judge, as a text)
[Judge] (your judge, as a number, 0, 1, or 2)

## Note ##
The question is being asked by a user who is playing with a take-apart
    toy.
Gold answer(s) are created by well-trained humans.
Predicted answer is created by a machine, based on the corresponding
    instruction and the frames of the assemblying process recording.

## Task ##
Now, here are the question, gold answer(s), and predicted answer:
[Question]
{question}
[Gold Answer(s)]
{gold_answer}
[Predicted Answer]
{predicted_answer}

## Feedback ##
[Rationale]
```

Figure 12: LLM-as-a-judge prompt

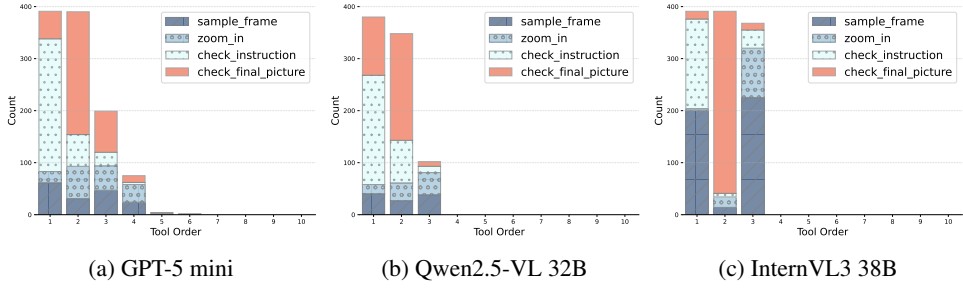

(a) GPT-5 mini      (b) Qwen2.5-VL 32B      (c) InternVL3 38B

Figure 13: Tool usage pattern for the remaining models.