# OpenReview forum: "TAMA: Tool-Augmented Multimodal Agent for Procedural Activity Understanding"
_ICLR.cc/2026/Conference — ICLR 2026 Conference Withdrawn Submission_

### Official Review · Reviewer_RWDR · 2025-10-28

**Soundness:** 2
**Presentation:** 3
**Contribution:** 2
**Rating:** 2
**Confidence:** 4

**Summary:**

The paper introduces TAMA, a training-free tool-augmented multimodal agent for procedural activity understanding. Unlike prior agent frameworks that convert vision to text and reason purely in language, TAMA runs a VLM as the agent and calls multimedia-returning tools (sample frames, zoom, check instructions, check final image) to interleave visual and textual reasoning. Evaluated on ProMQA-Assembly, TAMA often improves over naive single-pass, prompted reasoning, and a strong training-free workflow baseline (TCoT).

**Strengths:**

1. The paper targets real procedural tasks (assembly) and articulates why interleaved visual reasoning matters beyond text grounding.

2. Treating the VLM as the agent and preserving image modality via multimedia-return tools avoids captioning bottlenecks and is a meaningful shift from text-only agents.

3. Comparisons across proprietary/open-weight models; baselines include naive, reasoning, and TCoT; ablations on text vs multimedia tools and workflow vs agentic (plus presampling) substantiate claims.

4. Tool-use statistics and patterns illuminate why some models (e.g., GPT-5) benefit more, offering actionable insight for future training

**Weaknesses:**

1. Ad-hoc design / limited theory. Key choices (tool inventory, call policy, cut-ins, stopping) are heuristic. A lightweight formalization (e.g., VOI-regularized sequential acquisition / POMDP view) and a principled stopping rule would upgrade the contribution from an effective recipe to a grounded method.

2. Single-dataset scope. Results are only on ProMQA-Assembly; generality to other procedural domains remains untested

3. Real-world realism. Tools are idealized Python functions over local assets; discussion of latency, retrieval noise, and integration constraints is brief.

3. Judge-based evaluation. Reliance on LLM-as-a-judge could be complemented with targeted human evals or task-specific metrics to validate assistance quality.

**Questions:**

1. The authors are expected to add a theory sketch, eg., formalize the loop as sequential evidence acquisition with an accuracy–cost objective.

2. How sensitive is TAMA to prompt templates/tool naming, and do small perturbations change tool-use behavior or accuracy?

3. The authors are expected to provide failure analyses and per-tool VOI diagnostics to clarify when zoom vs sampling vs instruction checks help (building on Table 4 / Fig. 2).

---

### Official Review · Reviewer_mWJb · 2025-10-30

**Soundness:** 2
**Presentation:** 3
**Contribution:** 2
**Rating:** 4
**Confidence:** 3

**Summary:**

TAMA is a training-free, agentic framework that lets a vision-language model interleave reasoning with simple, perceptual tools that return information in its native modality—uniform frame sampling, zoom-in cropping, instruction lookup, and final-picture lookup—so the model can “think with images” while answering procedural video questions. Evaluated zero-shot on ProMQA-Assembly against (i) a naive concat baseline, (ii) text-only reasoning, and (iii) a two-stage workflow, TAMA generally raises accuracy—most notably for GPT-5, GPT-5 mini, MiMo-VL 7B, and Gemini 2.5 Flash—while yielding little or no gain for Qwen2.5-VL 32B and InternVL3 38B and trailing text-only reasoning slightly for Claude 4 Sonnet. Ablations show that returning images beats caption-only tool outputs, an agentic loop is more robust than hand-designed workflows (with one tie for Gemini), and light presampling of frames helps some models. Cost analyses indicate TAMA is often cheaper than TCoT at comparable or better accuracy, and usage logs highlight frequent reliance on the zoom-in tool. Limitations include occasional prompt “cut-ins” to keep the loop on track and weaker adherence for some open-weight models, but overall the results suggest agentic, image-centric tool use is a strong path for procedural activity understanding.

**Strengths:**

TAMA’s training-free, agentic design makes it a drop-in control loop for many VLMs, and its “think-with-images” approach—using a small, clear toolset for frame sampling, zooming, instruction lookup, and final-picture retrieval—preserves crucial visual detail. The method generalizes across both closed- and open-weight models, and the paper provides solid empirical rigor with zero-shot baselines, agentic-vs-workflow comparisons, and targeted ablations that explain where gains come from. It is often cost-efficient relative to stronger workflows, while its explicit tool calls and visual crops improve interpretability and traceability. By focusing on realistic procedural activity understanding (e.g., assembly), the work is practically relevant, and its modular design is easy to extend without retraining, supported by a clear benchmark and evaluation protocol that facilitate reproducibility.

**Weaknesses:**

Gains are not universal—TAMA trails a strong workflow baseline for Gemini, Claude’s text-only reasoning slightly outperforms it, and open-weight models like Qwen2.5-VL 32B and InternVL3 38B show little or no improvement, suggesting limited robustness across models. The agent loop required heuristic “cut-in” prompts to steer behavior, and some open-weight models struggled to follow ReAct-style interleaving, implying extra tuning. The evaluation leans on single-run, LLM-as-a-judge scoring, which invites variance and judge bias, and API/template instability makes cost and reproducibility comparisons murkier. Finally, the study is scoped to ProMQA-Assembly, so external validity to other tasks and datasets remains to be shown.

**Questions:**

1. How do scores vary with prompt templates, API versions, and decoding settings across all models?
2. Does TAMA transfer to other procedural domains/datasets (e.g., ProMQA-cooking, Ego4D, HowTo) without retuning?

---

### Official Review · Reviewer_f7H5 · 2025-11-01

**Soundness:** 3
**Presentation:** 3
**Contribution:** 2
**Rating:** 2
**Confidence:** 4

**Summary:**

This paper proposes TAMA (Tool-Augmented Multimodal Agent), a novel framework designed for procedural activity understanding. Its primary contribution is a training-free agentic framework that enables a Vision-Language Model (VLM) to perform interleaved multimodal reasoning by flexibly using tools that return multimedia (images or text) rather than just text. The experiments demonstrate that this approach can improve the performance of specific VLMs, notably GPT-5 and MiMo-VL, on the ProMQA-Assembly dataset. Finally, ablation studies provide empirical support validating the framework's two core features: the benefits of multimedia-returning tools over text-only tools and the effectiveness of agentic, flexible tool selection over fixed workflows.

**Strengths:**

- The paper is well written and easy to follow.
- The method is simple and easy to understand.

**Weaknesses:**

- The paper makes broad claims about "procedural activity understanding" and "video-centric multimodal tasks", but the experimental validation is confined to a single dataset and a limited, image-centric toolset.
- The paper states that open-weight models "sometimes failed to follow the intended ReAct-style prompting" and "occasionally refused to output any reasoning". This is an observation of the failure, not an analysis of its cause. Why did they fail? Did they get stuck in loops? Call non-existent tools? Ignore the tool outputs?
- This paper's contribution is more focused on prompt engineering, but the designed system is rather simplistic (a 'toy' system) and the validation scope is very narrow. It is recommended that the authors validate the framework's capabilities on more datasets.

**Questions:**

See weaknesses.

---

### Official Review · Reviewer_tf8c · 2025-11-03

**Soundness:** 3
**Presentation:** 3
**Contribution:** 3
**Rating:** 6
**Confidence:** 3

**Summary:**

The paper introduces TAMA, a training-free, tool-augmented multimodal agent framework designed for procedural activity understanding, such as cooking, assembly, and lab experiments. Unlike conventional single-pass or workflow-based video understanding methods, TAMA leverages interleaved multimodal reasoning by allowing a vision-language model (VLM) to proactively call multimedia-returning tools (e.g., frame sampling, zooming, checking instructions, and final product images).
The model alternates between generating “thoughts” and executing these tool calls to iteratively refine its understanding of video content. Experiments on the ProMQA-Assembly dataset show that TAMA improves performance over naive, reasoning, and Temporal-CoT baselines for several models (notably GPT-5 and MiMo-VL). Ablation studies further confirm the value of multimedia-returning tools and agentic, flexible tool selection.

**Strengths:**

- The idea of training-free multimodal reasoning with tool interaction is original and timely, aligning with the emerging “thinking-with-images” paradigm.
- The evaluation includes both proprietary (GPT-5, Claude 4, Gemini) and open-weight (Qwen2.5-VL, InternVL3, MiMo-VL) models, providing a broad and fair comparison.
- Consistent improvements on multiple models (especially GPT-5 and MiMo-VL) show the practical benefit of interleaved multimodal reasoning even without additional training.

**Weaknesses:**

- The evaluation focuses solely on ProMQA-Assembly; results on other procedural datasets (e.g., cooking or lab tasks) would strengthen generalization claims.
- Some open-weight models (Qwen2.5-VL, InternVL3) underperform under TAMA, suggesting the framework is not yet robust across architectures.
- The work is more of a system-level demonstration than a theoretically grounded method; clearer explanation of why interleaved reasoning helps would enhance its contribution.
- Using an LLM-as-a-judge for scoring introduces subjectivity; additional human or quantitative metrics would improve reliability.

**Questions:**

- How sensitive is TAMA’s performance to the design of tool APIs (e.g., parameter choices in frame sampling or zoom-in ranges)?
- Could the proposed agentic framework be extended to training-based adaptation, where models learn to select tools more efficiently?
- How does TAMA perform on longer or more complex videos, where reasoning depth and memory become critical?

---

### Note · Authors · 2025-12-03

**Comment:**

We thank all the reviewers for their constructive and thoughtful reviews. As suggested by reviewers, we evaluated our framework on other datasets, and we found a lack of robustness of our framework across multiple datasets. Hence, we withdraw our paper and investigate more.

**Withdrawal Confirmation:**

I have read and agree with the venue's withdrawal policy on behalf of myself and my co-authors.